# Apparent Pyrolysis Kinetics and Index-Based Assessment of Pretreated Peach Seeds

Angelos-Ikaros Altantzis [ID], Nikolaos-Christos Kallistridis [ID], George Stavropoulos and Anastasia Zabaniotou *[ID]

Department of Chemical Engineering, Aristotle University of Thessaloniki, 54124 Thessaloniki, Greece; ikaros.altantzis@gmail.com (A.-I.A.); nkallist@cheng.auth.gr (N.-C.K.); gstavrop@auth.gr (G.S.)
* Correspondence: azampani@auth.gr; Tel.: +30-694-599-0604

**Abstract:** To better understand pyrolysis for upscaling purposes, a kinetic characterization of the process is necessary for every feedstock. Laboratory experiments allow identification of apparent kinetic models. This paper aims at the apparent kinetic investigation of peach seeds' slow pyrolysis. Peach seeds from Greek peach fruits pyrolyzed under inert atmospheric conditions at different temperatures (475–785 °C), heating rates (100–250 °C/min) and $N_2$ flow rates (25–200 cc/min). Prior to pyrolysis, they submitted to hexane extraction for the recovery of 36.8% wt. of the contained oils. Determination of the specific rate constant (k) and activation energy (Ea) for each considered reaction was made by using the Coats–Redfern integral non-isothermal fitting model that requires an assumption of the reaction order (n). Results revealed that a 3rd order reaction model best fits the process, the increasing of the pyrolysis temperature leads to a decrease of the activation energy (E) and pre-exponential factor (A), while nitrogen flow rate and heating rate had an opposite impact. E and A values ranged from 23 to 56 kJ/mol and $1.82 \times 10^6$ to $1.13 \times 10^6$ min$^{-1}$, respectively, at different pyrolysis conditions. Furthermore, estimation of combustion and pyrolysis indexes were made to assess the suitability of peach seeds as a fuel, using isothermal thermogravimetric analyses (TGA). Results revealed that peach seeds are a suitable feedstock for pyrolysis, while prior submission of peach seeds to oils extraction, in a cascade biorefinery approach, can increase the energy and material recovery efficiency and potentially the environmental and economic benefit of the agri-food industrial sector.

**Keywords:** pyrolysis; peach seeds; kinetics; apparent; index; TGA/DTG; Coats-Redfern method

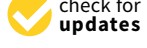



## 1. Introduction

Pyrolysis is a promising biorefinery technology for energy and resources efficiency in the Circular Bioeconomy. Pyrolysis of biomass is a thermochemical process that displays numerous benefits. It is a method of closing energy and materials loops, while decreasing the volume of residues generated, and improving the economic viability of the industrial sector, if the produced bioenergy and/or biochar is used in-site [1,2]. It is quite friendly for the environment, since it produces fuels that do not burden the atmosphere with pollutants ($SO_X$, $NO_X$) due to the low content in nitrogen and sulfur into the biomass primitive form. Furthermore, considering the bioresource development through photosynthesis, the total $CO_2$ footprint in the environment is negligible [3].

During the last decades, the international scientific community has been focused on the pyrolysis of various types of biomass to understand the process mechanisms and determine the kinetic parameters of the main reactions. Fast pyrolysis and the searching for appropriate catalysts for enhancing the derived bio-oil yields were in the center of researchers' focus because liquid products were the main interest as transportation biofuels [1–4].

Meanwhile, the challenges of limited resources availability and climate change have triggered the development of the food waste biorefinery concept. Increased demand for food production is resulting in the generation of agri-food wastes that are seen as the

key strategy for the development of sustainable industrial processes via several environmentally friendly technologies to obtain products with high added value to supply the chemical, pharmaceutical and food and energy industries [4]. The valorization of agricultural processing wastes and agri-food residues generated by the agro-industrial business could be an effort to mitigate environmental impact by their transformation thought a biorefinery approach, also integrating the pyrolysis process for bioenergy and biochar production towards zero-waste solutions [2]. Waste biorefinery is a sustainable practice for increasing sustainability through circularity and resource efficiency in a circular bioeconomy, for producing multiple products (biobased products, bioenergy and fuels) from various biomass feedstocks through the incorporation of relevant conversion technologies, including pyrolysis [5–7].

Fruit production holds an important share in the agricultural production of many countries and of the Greek agricultural economy. It constitutes a vital factor in the commercial balance of agricultural products, while peach trees are the most widely cultivated fruit trees in the Central Macedonia region of Greece. Peach and other fruits are processed by the food and beverage agro-industrial companies in the region to produce various juices, sweets, etc., while stones, husks, kernels, seeds remain as residues in important amounts [8–10].

Towards the development of new agro-industrial processes with social eco-innovation benefits, valorization of tree fruit stone and seed wastes in the concept of waste biorefinery is an ecologically sustainable solution that benefits from material and energy recovery and closed loops. Biorefining the inedible part of the tree fruits (stones, husks, kernels, seeds) that constitutes a huge portion of the fruit processing agro-industrial solid waste is challenging for the sector, considering the diversity of fruits, their cultivators' heterogeneity and seasonal production [10].

Towards designing an efficient pyrolysis process of fruits seeds and scale up of reactors for energy production, a pyrolysis parametric investigation for each specific feedstock is required, along with simplified kinetic modeling. In the international literature, a plethora of papers for numerous lignocellulosic materials can be found on parametric pyrolysis estimation and comparison of the optimum values of the important kinetic parameters affecting the pyrolysis process [11–16]. An important number of papers on kinetic modeling of various biomass segments have reported kinetic methods, categorized as model-free and model-fitting methods [17,18]. The most common papers explored are the model-free analysis, based on the iso conversional methods to obtain kinetic parameters from TGA results, providing a preliminary understanding of the reactions, preliminary results and often involving assumptions of the numerical approximations and complex conversion data [19]. Other researchers have investigated model fitting methods, by using distributed activation energy models and simplified multi-step kinetic models with varied reaction schemes (multi-step modeling methods). The first reported model of this type was the Broido–Shafizadeh model for cellulose pyrolysis at low temperature, shown in the international literature in 1995 [20], which was modified later [21]. Multi-component devolatilization mechanisms to address the chemical complexity and heterogeneity of the biomass content have been proposed by other authors who adopted the distributed activation energy model (DAEM) [19]. Burra and Gupta (2019), by adopting a distributed activation energy model (DAEM), proposed a simplified kinetic model including three independent parallel, competitive and successive reactions for biomass pyrolysis. In this model, one or several irreversible first order parallel reactions were used with invariant pre-exponential factor and a continuous distribution function (usually Gaussian), representing the activation energy. By using the sequential multi-step reaction model and simultaneous thermal analyses (TGA-DSC), they further investigated the pyrolysis of different kinds of unconventional lignocellulosic biomass materials (cellulose, cardboard, paper waste, pinewood, rice husk and chicken manure) revealing that the lignocellulosic materials require at least ~3-step sequential model for accurate pyrolysis modeling [19]. Pseudo-first order models, parallel, successive and competitive reaction schemes and complex reaction networks for cellulose,

hemicellulose and lignin were also used [22]. Researchers have modeled lignocellulosic materials fast pyrolysis based on the apparent kinetics [23]. Contrary to non-isothermal models, the isothermal models for the estimation of kinetic parameters are more advantageous because they are not influenced by the specific mathematical model used [24].

The published works on peach pyrolysis and kinetics are very few, mainly based on TGA/DTGA experiments. Mathematical modeling by using a non-isothermal model, built on data gathered from TGA/DGTA procedures, has been investigated by two research groups from Italy, Spain, and Serbia [25–28]. A devolatilization kinetics through combined application of model-free methods and DAEM (distributed activation energy model) using Gaussian distribution functions of activation energies was also studied for peach seeds by the Serbian research team [29].

Apparent kinetic studies and practical models that predict the evolution of specific products of each organic waste undergoing pyrolysis are needed towards upscaling the process, because pyrolysis is a process that cannot be strictly described by a simple mechanism, since it involves multiple complex and antagonistic reactions [30,31].

*Scope and Objective*

There is a need to investigate peach seeds pyrolysis kinetics, despite the plethora of studies on various food waste pyrolysis, since only few papers on peach seeds from different cultivars can be found in the literature. A full kinetic characterization of the pyrolysis process complexity on a mechanistic basis can be useful in understanding the process and its variables, in predicting the process dynamics under variable operating conditions. Therefore, this study aims to investigate the apparent kinetics of peach seeds pyrolysis from peach trees cultivated in Greece, with the aim to provide a tool supporting the pyrolysis process optimization and scale-up.

The research objective of this paper was four-fold:

(1) Pretreatment of peach seeds by hexane to recover the oils, in the biorefinery approach.
(2) Pyrolysis of the remaining solid, after extraction of lipids, in a laboratory plug flow reactor (PFR) for the parametric study of the effect of temperature, heating rate and the carrier gas flow on the pyrolysis products yield.
(3) Kinetic study of the process by using the non-isothermal model proposed by Coats and Redfern fitting the model [32] on the experimental results to estimate the reaction order (n), specific rate constant (k) and activation energy (Ea).
(4) Estimation of quality indexes based on isothermal TGA data for assessing the appropriateness of the pretreated peach seeds for the pyrolysis process.

The study provides information on the apparent kinetic modeling of pretreated peach seeds pyrolysis contributing to the knowledge required for the optimization of the pyrolysis process efficiency in the transition to a circular economy. Valorization of the agri-food residues and waste constitute an important sector within the agricultural production of Greece, holding a very important share in the domestic agricultural economy, with peach production to constitute a vital agricultural sector in the northern country. Canneries and juice processors in Greece generate annually >20,000 tn of solid waste from peaches, apricots and cherries processing [10]. Valorization of peach seed via cascade oil extraction and pyrolysis of the remaining solid could be an ecologically sustainable solution with material and energy savings in the circular bioeconomy journey that the country is undertaking.

## 2. Materials and Methods

To determine the best way to lump a kinetic model, identification of experimental data and numerical rules must be applied. For estimating the kinetic parameter values, several experimental-numerical techniques can help in pointing out the process parameters' interaction.

### 2.1. The Material-Feedstock

Samples of peach seeds were provided by the Greek Canners Association [33]. These solid residues were milled to a particle size < 1 mm, and dried in an air dryer at 110–120 °C for at least 48 h. Then, experiments were performed. Prior to pyrolysis, peach seeds have undergone hexane extraction, aiming to first extract the oils and then pyrolyze the remaining solid residue, in the concept of a waste cascade biorefinery in the circular bioeconomy. The pretreated peach seeds were stored under low temperatures of (−20 °C) to avoid any deteriorations through microorganism development, before fueling the pyrolysis reactor.

### 2.2. Protocol of Feedstock Extraction with Hexane

Peach seeds contain oil and bioactive compounds, exceeding apricot and sour cherry in terms of oil yield (48% vs. 38% and 26%, respectively). Peach seed oils are considered rich in monounsaturated fatty acids, mainly oleic acid (43.9–78.5%), moderate sources of linoleic acid (9.7–37%) and palmitic (4.9–7.3%) ones [10].

The extraction process with MERCK hexane was performed. First, there was mechanical separation of the wooden pit (endocarp) from the removed peach kernel and the collected sample was preheated at 40 °C for 30 min. After grinding and weighing, the sample was transferred to a conical flask where hexane was added (as an extraction solvent) in a ratio of 1:3 $w/v$. Extraction occurred for 3 h in an ultrasonic bath and environmental temperature. Liquid phase included the oils and the solvent which removed via a vacuum filtration and separated from the solid filtrate. The solid residue was dried for at least 48 h under temperatures of 110–120 °C.

Pretreated peach seeds samples were stored under low temperatures of −20 °C, in order to avoid any deterioration through microorganism development, prior of their use in pyrolysis.

The extraction of oils yielded 36.98% wt.

### 2.3. Pyrolysis Experimentation Protocol

For the apparent kinetics estimation, sets of pyrolysis experiments were conducted, and the mass and volume ratios used for the kinetic study were estimated. The dynamic experimental procedure (temperatures, flows) was recorded, and the produced database of each experiment was used from the initial until final peak temperature where devolatilization occurred. The integral method proposed by Coats and Redfern was used for the estimation of apparent kinetic parameters.

The sample was dried, pulverized and sieved to a particle size < 1 mm prior to placement within the plug flow reactor (PFR) at a defined heating rate, where nitrogen flowed for 15 min, while the temperature rise was recording. An initial sample mass of approx. 1 $gr$ of spherical shaped granules with a particle size varying between 500–850 μm was evenly arranged in a metal plate and placed in the reactor's center. In the initial sample, moisture did not exceed 10% wt./wt. Nitrogen inlet at a typical flow rate of 100 cm³/min inside the reactor was regulated at ~300 kPa. Inlet pressure should not be exceeded any further, since, in combination with the rapid product release, a burst in the reactor caps could take place. The products were collected, and their yield was calculated through analysis of the end products.

The monitoring parameters in each experiment were the heating rate and the total volume of the products. They were monitored through a dynamic recording of the whole experimental procedure (temperature and flow vs. time) by using the Advantech Genie Runtime® software (version 3.0x). For the temperature monitoring purposes, two thermocouples were used, the first placed in the exterior of the reactor, and the second just above the center of the sample.

The estimation of the solid product yields was done by comparing the initial sample mass with the final mass of the biochar (pyrolytic solid product) while the liquid products were measured simply by weighing the obtained oils in the tubes after quenching. Finally, gas product yields were estimated by extraction of the above measured masses from the initial sample mass.

### 2.4. Thermogravimetric Analysis (TGA/DTGA)

TGA/DTG analysis was carried out for the determination of the combustion and pyrolysis indexes that were used to assess the suitability of peach seeds as feedstock for pyrolysis.

A TG 209 F3 Tarsus® analyzer was used. The analyses were performed in air atmosphere with an air flow of 20 cm$^3$/min, heating rate of 20 °C/min and maximum temperature of 700 °C. The samples used were of an initial mass of approx. 1 mg. The pretreated peach seeds and the pyrolytic solid residue (biochar) produced under the reference conditions ($N_2$ flow rate = 100 cm$^3$/min, heating rate = 170 °C/min and maximum temperature = 650 °C) were used for the TGA/DTGA experiments.

For the processing of the results, the Proteus® computer software was used [34].

### 2.5. Apparent Pyrolysis Kinetics

Biomass pyrolysis, due to the range and complexity of the reactions that govern it, is not possible to be described through a single mechanism with absolute accuracy. It can, however, be approximately described with sufficient precision, by one or more kinetic models proposed in the literature [35]. During the thermal decomposition of biomass, many complex and antagonistic reactions occur simultaneously. This is the reason why the exact mechanism of pyrolysis remains unknown and there are many models of kinetic equations in the literature to describe it.

Reaction dynamics and kinetic equations are influenced by three main factors. These are the breaking and redistribution of chemical bonds of molecules, the changes in the geometry of reactions, and the two-phasic diffusion between reactants and products [36].

As the exact mechanism by which cracking reactions in pyrolysis take place is not known, a general approach to the phenomenon is taken through kinetic models of varying degrees of difficulty and accuracy. All these models are the result of research by the scientific community and are based on an original approach developed by Homer Kissinger in the mid-1960s, based on which the activation energy and the kinetic parameters for the thermal processes can be calculated for thermal processes [37].

The models can be categorized based on the temperature profile prevailing in the reactor, which can be either isothermal or non-isothermal. The parameter that determines the appropriate temperature profile is the heating rate. Therefore, if the heating rate is particularly high, an isothermal profile is presumed. A common denominator on which all kinetic models are based is the assumption that pyrolysis is the decomposition of functional groups from the carbon structure through a series of independent and non-independent reactions [34–38].

In the current study, pyrolysis heating rates of 100–250 °C/min were applied; therefore, a non-isothermal profile is used to describe the devolatilization kinetics described by the Equation (1) [36]:

$$-\frac{dV}{dt} = k(V_0 - V)^n,\tag{1}$$

where:

$n$ = Order of reaction.
$V$ = Volume of gases in time t per gram of carbon, in scm$^3$gr$^{-1}$.
$V_0$ = Volume of gases in total devolatilization per gram of carbon, in scm$^3$gr$^{-1}$.
$k$ = Reaction rate constant of Arrhenius equation, given by the Equation (2).

$$k = Ae^{(-\frac{E}{RT})},\tag{2}$$

where:

$A$ = Pre-exponential factor, in sec$^{-1}$ or min$^{-1}$.
$E$ = Activation energy, in Jmol$^{-1}$.
$R$ = Universal gas constant (8.314 Jmol$^{-1}$K$^{-1}$).
$T$ = Absolute temperature, in K.

By using the heating rate equation: $q = \frac{dT}{dt} = constant \neq 0$, the Equation (1) gives the Equation (3):

$$\frac{dV}{dT} = \left\{ \left( \frac{k_0 V_0}{q} \right) e^{\left( -\frac{E}{RT} \right)} \left[ V_0^{1-n} \frac{k_0}{q} I(T) \right] \right\}^{\frac{n}{1-n}},\tag{3}$$

where:

$$I(T) = \int_{T_0}^{T} -e^{\left( -\frac{E}{RT} \right)} dT,\tag{4}$$

Afterwards, integrating the Equation (4) leads to the Equation (5):

$$V = V_0 - \frac{1}{\left[ \left( \frac{1}{V_0} \right) + \left( \frac{k_0}{q} \right) \right] I(T)},\tag{5}$$

The kinetic parameters $k_0$, E are calculated by using the graphical representation of the Equation (6):

$$ln \left[ \frac{q}{(V_0 - V)^n} \frac{dV}{dT} \right] = \ln(k_0) - \frac{E}{R} \frac{1}{T},\tag{6}$$

The calculation of the parameter $k_0$ is based on the y-intercept of the straight line, while the slope of the line $(-E/R)$ is equal to the activation energy, E.

It must be noted that this specific kinetic model is a simplified approach to the phenomenon of pyrolysis and does not describe with absolute accuracy the rate of release of volatile chemical compounds.

The general form of the above equation for the case of non-isothermal operation can also be written by the Equation (7), [38]:

$$ln \left[ \frac{g(a)}{T^2} \right] = ln \left( \frac{AR}{qE} \right) - \frac{E}{R} \frac{1}{T},\tag{7}$$

where:

$$g(a) = \int_0^a \frac{da}{f(a)},\tag{8}$$

$f(a)$ = Kinetic function dependent on the prevailing mechanism.
$a$ = Pyrolysis conversion rate of sample, given by the Equation (9).

$$a = \frac{W_{initial} - W_i}{W_{initial} - W_{final}} = \frac{V_{initial} - V_i}{V_{initial} - V_{final}},\tag{9}$$

where:

$W_i$, $V_i$ = Actual sample mass and volume at current time, respectively.

From the graphical representation of $ln \left[ \frac{g(a)}{T^2} \right]$ against $1/T$, the slope $-E/R$ of the straight line can be determined, since $ln \left( \frac{AR}{qE} \right)$ is nearly constant, based on which, the activation energy is calculated.

In this study, the expressions shown in Table 1 were tested to calculate the activation energy needed for the release of the volatile products during the peach seeds pyrolysis.

**Table 1.** Common pyrolysis reaction mechanisms for solid state reactions [38].

| Rate-Determining Mechanism | Reaction Type | $f(a)$ | $g(a)$ |
|---|---|---|---|
| Chemical reaction | First-order | $1 - a$ | $-\ln(1 - a)$ |
| | Second-order | $(1 - a)^2$ | $(1 - a)^{-1} - 1$ |
| | Third-order | $(1 - a)^3$ | $[(1 - a)^{-2} - 1]/2$ |
| | nth-order | $(1 - a)^n$ | $[(1 - a)^{1-n} - 1]/(n - 1) - 1$ |
| Random nucleation and nuclei growth | Bi-dimensional | $2(1 - a)[-\ln(1 - a)]^{1/2}$ | $[-\ln(1 - a)]^{1/2}$ |
| | Three-dimensional | $3(1 - a)[-\ln(1 - a)]^{2/3}$ | $[-\ln(1 - a)]^{1/3}$ |

**Table 1.** *Cont.*

| Rate-Determining Mechanism | Reaction Type | $f(a)$ | $g(a)$ |
|---|---|---|---|
| Diffusion | One-way transport | $1/(2a)$ | $a^2$ |
| | Two-way transport | $[-\ln(1-a)]^{-1}$ | $a + (1-a)\ln(1-a)$ |
| | Ginstling–Brounshtein equation | $(2/3)(1-a)^{2/3} - (1-a)^{1/3}$ | $1 - 2a/3 - (1-a)^{2/3}$ |

The reactions that occur during a pyrolysis process are distinguished in three main stages. Initially, at temperatures < 250 °C, moisture entrapped within the biomass structure is vaporized and removed, a process known as drying. As the temperature in the rector increases, primary and secondary reactions among pyrolysis products take place that ultimately determine both the qualitative and quantitative product composition. In a temperature range between 300 °C and 500 °C, formation of primary products takes place through a mechanism that involves mainly depolymerization (where bonds between monomers in the polymeric chain are cracked), fragmentation (where covalent bonds break thermally in different parts of the polymeric chain) as well as the primary char formation [39].

Volatile molecules produced through polymerization reactions have relatively high molecular weight, and thus, are easily condensed, forming in this way the primary liquid (tar) products [40]. The main products of this stage are illustrated in Figure 1.

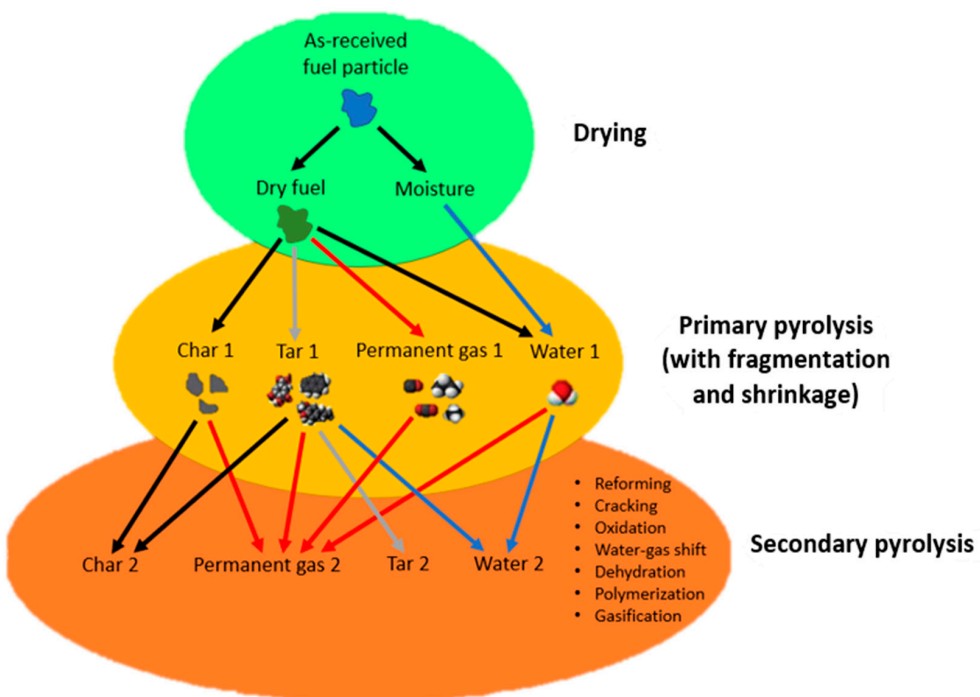

**Figure 1.** Schematic representation of pyrolysis reaction mechanism, adapted from Neves et al. (2011) [40].

When the reactor temperature increases even further, primary released products undergo further conversion both homogeneously and heterogeneously, leading to a secondary product release. More specifically, serial and parallel reactions take place at temperatures > 500 °C, such as cracking, dehydration, polymerization and gasification reactions.

The primary formed biochar could further undergo gasification reactions while simultaneously acting as a catalyst for cracking reactions among volatiles for lighter gas production, or either for repolymerization reactions leading in the formation of the secondary char. However, this categorization is not absolute, since primary and secondary reactions can occur simultaneously in different parts of the biomass feedstock.

Finally, the extent of these secondary reactions is the determinant factor for the final products spectrum and yields. As a result, the parameter effect in the product yields essentially refers to the degree of extent of secondary reactions.

### 2.6. Fuel Quality Indexes

Fuel quality indicators are related to the suitability of a residue/waste for thermal process. Indexes are practical and easily computable tools for comparing different fuels or even comparing fuels from pyrolysis of the same raw material, but under different conditions [41].

In the literature, there are many characteristic parameters which are used to characterize the combustion quality of liquid and solid fuels. All of these are based on experimental results derived from DGA/DGTA. The process involves combustion experiments for obtaining weight change curves, and combustion rates of the sample as a function of temperature. For the calculations, the combustion ignition temperature is needed, which is determined by the outset of weight drop in the thermogram. Furthermore, the burnout temperature is required, which is determined from the turning point to 99% conversion in the thermogram [42].

The calculation of the characteristic combustion index C related to the combustion rate of a fuel [43], is described as follows. The rate of combustion is proportional to the kinetic constant, which in turn depends on temperature through the Arrhenius equation. Thus, the rate of combustion of a solid (or even liquid) fuel, in terms of its temperature dependence, is written by the Equation (10), [44]:

$$\frac{dM}{dt} = A e^{-\frac{E}{RT}}, \tag{10}$$

where:

$M$ = Fuel mass, usually in kg.
$A$ = Pre-exponential factor, in $s^{-1}$ or $min^{-1}$.
$E$ = Activation energy, in $Jmol^{-1}$.
$R$ = Universal gas constant ($8.314\ Jmol^{-1}K^{-1}$).
$T$ = Temperature, in K.

Differentiating Equation (10) gives the Equation (11):

$$\frac{R}{E}\frac{d}{dT}\left(\frac{dM}{dt}\right) = \frac{dM}{dt}\frac{1}{T^2}, \tag{11}$$

Equation (11) is also valid for the Ignition temperature ($T = T_i$):

$$\frac{R}{E}\frac{d}{dT}\left(\frac{dM}{dt}\right)_{T=T_i} = \left(\frac{dM}{dt}\right)_{T=T_i}\frac{1}{T_i^2}, \tag{12}$$

Based on Equation (12) the characteristic fuel combustion index $C$ is defined by the Equation (13), as:

$$C = \frac{R}{E}\frac{d}{dT}\left(\frac{dM}{dt}\right)_{T=T_i}\frac{\left(\frac{dM}{dt}\right)_{max}}{\left(\frac{dM}{dt}\right)_{T=T_i}}\frac{\left(\frac{dM}{dt}\right)_{mean}}{T_e}Q, \tag{13}$$

where:

$T_i$ = Ignition temperature, in K.
$T_e$ = Burnout temperature, in K.
$Q$ = Released energy during fuel combustion, in Joule.
$\left(\frac{dM}{dt}\right)_{max}$ = Maximum combustion rate, namely, rate at peak of DTG curve in mg/min or in % weight loss/min.

$\left(\frac{dM}{dt}\right)_{mean}$ = Average combustion rate, namely, half the value at the peak of the DTG curve in mg/min or in % weight loss/min.

It is obvious that the higher the value of index *C*, the better are the combustion characteristics of the fuel. In the Equation (13) of index *C*, the *R/E* ratio, as mentioned before, is directly related to the combustion rate. The higher the activation energy, the lower is the reactivity of the fuel.

The terms included in the equation of index *C* are the following [44]:

$$\frac{d}{dT}\left(\frac{dM}{dt}\right)_{T=T_i} \tag{14}$$

The term (14) refers to the change in combustion rate at the ignition temperature and is directly related to the ignition temperature. High values of this term indicate more intense ignition of the fuel [44].

$$\frac{\left(\frac{dM}{dt}\right)_{max}}{\left(\frac{dM}{dt}\right)_{T=T_i}} \tag{15}$$

The above term (15) expresses the ratio of maximum combustion rate to the combustion rate at ignition temperature, that positively affects the combustion of the solid. The higher its value, the more intensely the fuel burns.

$$\frac{\left(\frac{dM}{dt}\right)_{mean}}{T_e} \tag{16}$$

The term (16) expresses the ratio of the average combustion rate to the combustion completion temperature and is the inverse of the combustion time. Its high values are an indication of short combustion times.

Substituting Equation (12) into Equation (13) gives the Equation (17):

$$C = \frac{\left(\frac{dM}{dt}\right)_{max}\left(\frac{dM}{dt}\right)_{mean}Q}{T_i^2 T_e}, \tag{17}$$

Besides the index *C*, some other indexes have been proposed [45]. These are the following: Ignition Index $D_i$ [45].

$$D_i = \frac{DTG_{max}}{T_i T_e} \tag{18}$$

where $DTG_{max}$ is the maximum combustion rate.

Flammability Index $C_{fi}$ [45].

$$C_{fi} = \frac{\left(\frac{dW}{dt}\right)_{max}}{T_i^2} \tag{19}$$

where $\left(\frac{dW}{dt}\right)_{max}$ describes the maximum combustion rate.

Combustion Stability Index $H_F$ [45], that indicates the stability of the combustion rate after ignition.

$$H_f = T_{max}ln\left(\frac{DT}{DTG_{max}}\right)10^{-3} \tag{20}$$

where:

$T_{max}$ = the maximum combustion rate corresponding temperature.
$DT$ = the exothermic peak width.
$DTG_{max}$ = the maximum value of DTG.

Similar quality indexes are used for pyrolysis. The pyrolysis index *D* expressing the volatile release is defined by the Equation (21) [46].

$$D = \frac{\left(\frac{dw}{dT}\right)_{max}\left(\frac{dw}{dT}\right)_{min}}{T_s T_{max,w} \Delta T_{1/2}},$$  (21)

where:

$\left(\frac{dw}{dt}\right)_{max}$ = maximum weight loss rate.

$\left(\frac{dw}{dt}\right)_{mean}$ = mean weight loss rate.

$T_s$ = starting temperature for volatile release and weight loss.

$T_{max,w}$ = temperature of maximum weight loss rate.

$\Delta T_{1/2}$ = temperature of full width at half maximum for the main peak of DTG curves.

## 3. Results

### 3.1. Experimental Results Used in the Kinetic Study

The elemental analysis of pretreated peach seeds is presenting in Table 2.

**Table 2.** Elemental analyses of pretreated peach seeds.

| Composition % wt./wt. | Pretreated Peach Seeds |
|---|---|
| C | 52.4 |
| H | 6.46 |
| N | 6.18 |
| S | 0.00 |
| O | 18.94 |
| Ash | 16.02 |

The experimental results used in the kinetic study are depicted in Table 3. These are the results of the pyrolysis parametric study related to the effect of the maximum pyrolysis temperature ($T_{max}$), the flow of the carrier gas ($F_{N2}$) and the heating rate (q) on the pyrolysis products' yield. In every pyrolysis experiment, the value of only one of these three conditions was changed to find the effect on the product yields.

**Table 3.** Effect of pyrolysis conditions on product yields.

| | $F_{N2}$ = 100 cm$^3$/min and q = 170 C/min. | | | |
|---|---|---|---|---|
| Experimental Set 1 | $T_{max}$ (°C) | 474 | 674 | 783 |
| | Biochar (gr) | 0.2644 | 0.2455 | 0.2313 |
| | Vapors (gr) | 0.7210 | 0.8076 | 0.7831 |
| | $T_{max}$ = 700 C, q = 170 C/min. | | | |
| Experimental Set 2 | Flow rate of $N_2$ (cc/min) | 25 | 50 | 100 | 200 |
| | Biochar (gr) | 0.2376 | 0.2353 | 0.2455 | 0.1556 |
| | Vapors (gr) | 0.7796 | 0.7793 | 0.8076 | 0.8923 |
| | $T_{max}$ = 700 K, $F_{N2}$ = 100 cm$^3$/min | | | |
| Experimental Set 3 | Heating rate, q (C/min) | 102 | 171 | 254 |
| | Biochar (gr) | 0.2586 | 0.2455 | 0.227 |
| | Vapors (gr) | 0.7540 | 0.8076 | 0.7905 |

### 3.2. Estimation of the Pyrolysis Reaction Order

After testing the various reaction expressions presented in Table 1, by fitting them to

the experimental pyrolysis results, the third-order chemical reaction was selected, because it describes the phenomenon most accurately by a straight-line equation (y = b × x + c).

### 3.3. Effect of the Pyrolysis Temperature on the Kinetic Parameters

Table 4 shows the results of the effect of the maximum pyrolysis temperature on the activation energy and the pre-exponential factor. These results are derived from the fitting of the experimental results in the model as shown in Figure 3. In Figure 3, the experimental data are colored points while the lines represent the simulated results from the third-order chemical reaction model.

**Table 4.** Effect of maximum pyrolysis temperature on the kinetic parameters E, A.

| $T_{max}$ (°C) | 474 | 674 | 783 |
|---|---|---|---|
| $R^2$ | 0.9888 | 0.9215 | 0.9520 |
| E (kJ/mol) | 56.7 | 31.5 | 23.3 |
| A ($min^{-1}$) | $1.34 \times 10^6$ | $1.82 \times 10^3$ | $0.13 \times 10^3$ |

E and A are calculated from the slope and the intercept of the equations of the straight lines (Figure 2). To assess the good fitting of experimental data with the simulated, the coefficient $R^2$ of the lines is used. $R^2$ have values in the range of 0.92 to 0.99. These values are very good, as in pyrolysis it is common to have a dispersion of results, indicating that the kinetic model simulates well the experimental results.

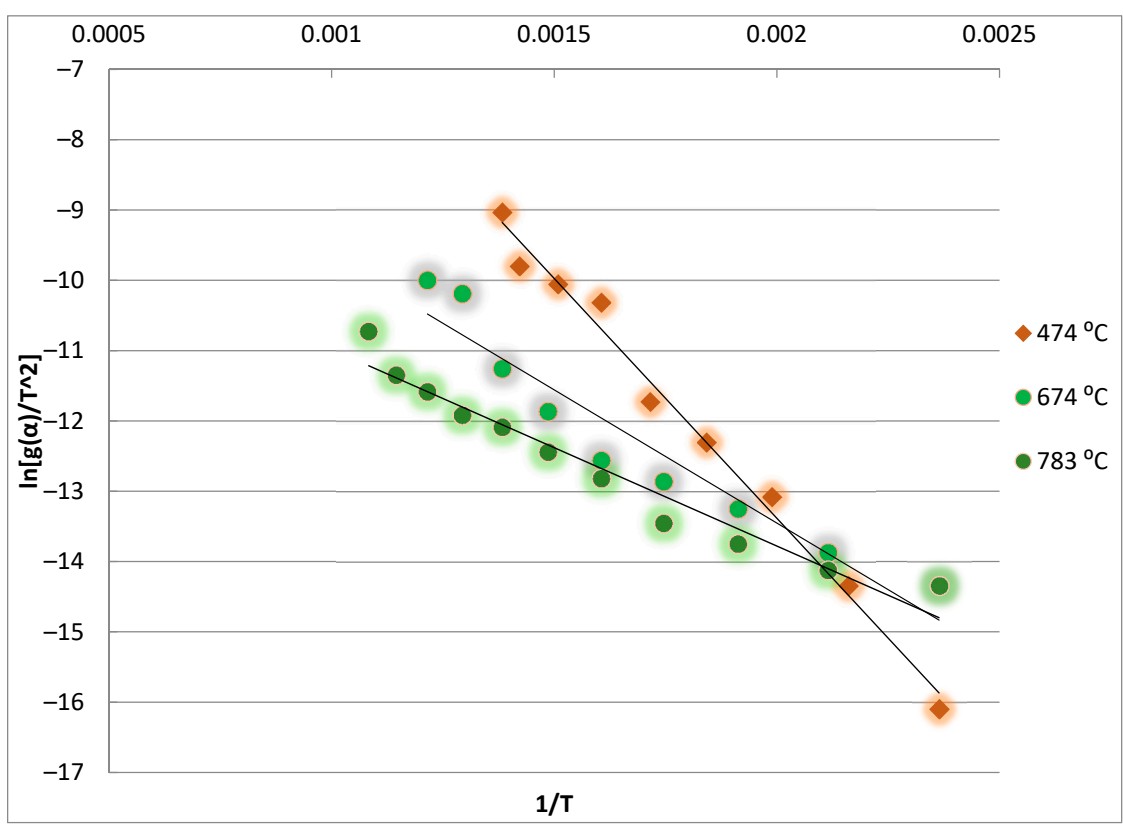

**Figure 2.** Effect of pyrolysis temperature on the kinetic parameters.

From Figure 2, it is obvious that by increasing the pyrolysis temperature, the slope and the ordinate of the kinetic equation decrease. Consequently, this leads to a decrease of the activation energy E and pre-exponential factor A, which have been presented in Table 4.

Based on the calculations performed, it appears that at the low pyrolysis temperature 474 °C, E is equal to 56.7 kJ/mol, while A is equal to $1.34 \times 106$ min$^{-1}$. As the pyrolysis temperature increases, both E and A decrease. Eventually, E decreases by about 60%, while A also decreases significantly. This is because increasing the $T_{max}$ favors fission-cracking reactions of loose bonded biomass components, which may be active groups on the outer surface of macromolecules, rather than the dissolution of molecules that are bound to the main mass of the raw material with multiple strong bonds. This can be attributed to the reduced rates of heat transfer to the inside of the particles due to the low thermal conductivity of the material. In addition, the decrease of the pre-exponential factor is due to the decreased cracking reactions rates. Since the reaction rate depends on E and A based on the Arrhenius equation, decreasing E and increasing T causes an increase in the heating rate, while decreasing A causes a decrease. Thus, these factors act as compensatory in regard to the heating rate. However, the effect of A prevails, since its reduction is stronger than E and T.

### 3.4. Effect of the Carrier Gas Flow Rate ($F_{N2}$) on the Kinetic Parameters

The effect of the carrier gas flow rate ($F_{N2}$) on the activation energy (E) and the pre-exponential factor (K) are depicted in the Table 5 and Figure 3.

**Table 5.** Effect of carrier gas flow (q) rate on kinetic parameters E, A.

| Flow Rate of N$_2$ ($F_{N2}$) (cc/min) | 25 | 50 | 100 | 200 |
|---|---|---|---|---|
| R$^2$ | 0.9902 | 0.9576 | 0.9215 | 0.9577 |
| E (kJ/mol) | 31.5 | 34.8 | 31.5 | 36.3 |
| A (min$^{-1}$) | $1.48 \times 10^3$ | $3.46 \times 10^3$ | $1.82 \times 10^3$ | $10.0 \times 10^3$ |

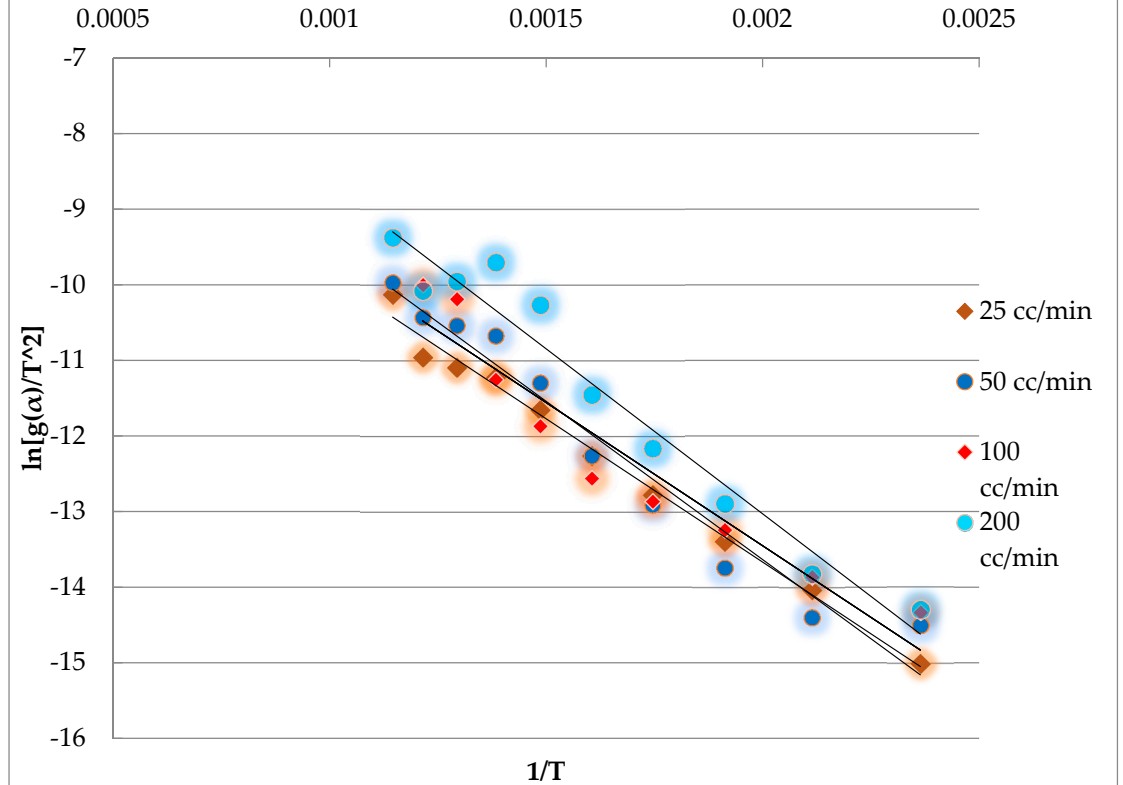

**Figure 3.** Effect of carrier gas flow rate on the kinetic parameters.

The flow rate of the carrier gas ($F_{N2}$) affects pyrolysis, since the change in flow rate causes a change in the residence time of the volatile products in the reaction zone and therefore, in the type of reactions that occur and the products that are produced. Four values of the flow rate were tested, ranging from 25 to 200 cm$^3$/min (Table 5). In all cases, third-order kinetic expressions were found to simulate the experimental results with high accuracy. The simulation is within acceptable limits since the value of the correlation coefficient ($R^2$) for all four cases ranges from 0.92 to 0.99, which is very good.

As can be noticed in Figure 4, the simulation lines are almost all parallel to each other, thus showing similar slopes values and therefore activation energies. Indeed, E shows small variations (Table 5) within the limits of the experimental error that can be considered as constant. The same applies to the first three values of the pre-exponential factor A, while at 200 cm$^3$/min, the value of A is increased fivefold. This might be due to the simulation, because A does not change significantly with the flow and remains always within the limits of experimental error. The apparent stability of E and A leads to the conclusion that the overall rate of cracking reactions is high enough to be completed in short residence times and thus, smaller $F_{N2}$. Moreover, the total activation energy of all the cracking reactions is mainly connected to the dissolution of the strongest bonds of the main mas of biomass. Besides, fission–reunion reactions have a relatively small contribution and many of them require small E. These small values of E are balanced by the exothermic reactions that take place in parallel with the secondary fission–reunion reactions. As a result, the secondary fission–reunion reactions have a small effect on the total activation energy of pyrolysis, as shown by the results of the present work (Table 5).

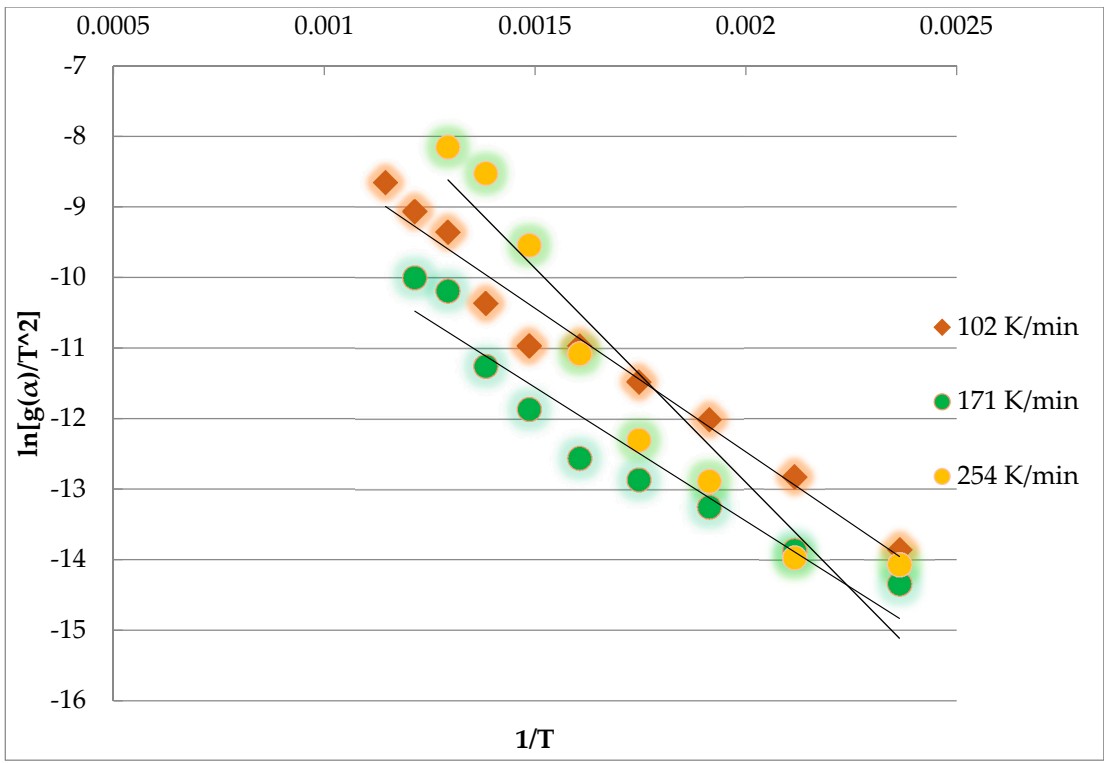

**Figure 4.** Simulation of kinetic model-Effect of heating rate.

### 3.5. Effect of the Heating Rate (q) on the Kinetic Parameters

The effect of the heating rate (q) on the kinetic parameters is shown in the Table 6 and Figure 4. The results of pyrolysis experiments at different heating rates and their simulation with third-order chemical reaction kinetics are presented in Table 6 and Figure 5, respectively. The correlation coefficients ($R^2$) are within acceptable limits, except that of the high heating

rate pyrolysis experiment (254 K/min), which is the only one with $R^2 < 0.92$ and much higher compared to the previous ones. (Table 6). The E and A values calculated by the model show small differences for heating rates of 102 K/min and 171 K/min and are 33.8 and 31.5 kJ/mol, respectively.

**Table 6.** Effect of heating rate (q) on the kinetic parameters E, A.

| Heating Rate, (q) (K/min) | 102 | 171 | 254 |
|:---:|:---:|:---:|:---:|
| $R^2$ | 0.9694 | 0.9215 | 0.9136 |
| E (kJ/mol) | 33.8 | 31.5 | 50.4 |
| A ($\text{min}^{-1}$) | $5.4 \times 10^3$ | $1.82 \times 10^3$ | $716 \times 10^3$ |

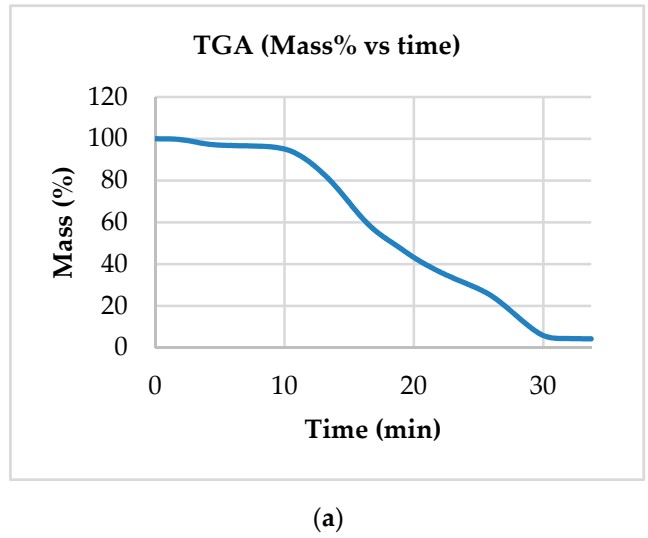

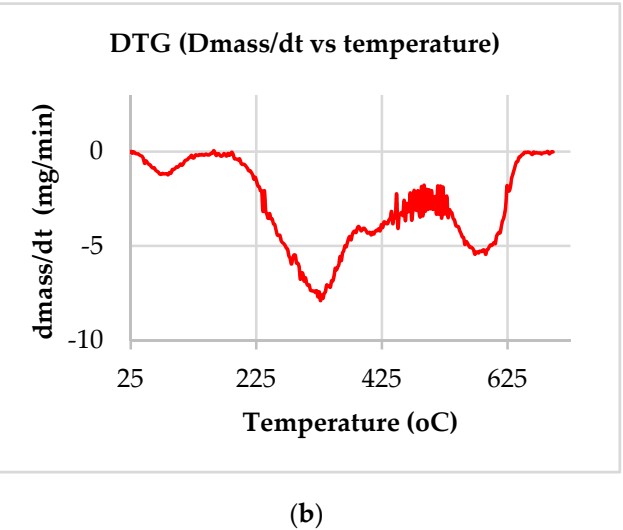

(**a**)

(**b**)

**Figure 5.** (**a**) Pretreated peach seeds weight loss diagram as a function of time (TGA, Air flow). (**b**) Pretreated peach seeds weight change rate diagram as a function of temperature (DTG, Air flow).

It can therefore be accepted with sufficient certainty that a significant increase in heating rate causes a significant increase in E and A. This conclusion is justified by the fact that a high heating rate causes faster heating of the biomass particles, allowing heat to penetrate throughout the mass of the granules, causing fission-cracking of all bonds of the material, which of course require large activation energies. At the same time, fission reactions are accelerated and that explains the high A value.

*3.6. Comparison of Peach Seeds Pyrolysis Kinetic Parameters with Bibliographic Data*

Under different pyrolysis conditions, the kinetic parameters are different as biomass is leaded to the breaking of different bonds resulting in the extraction of a slightly different kinetic equation each time.

In this study, E values ranged from 23–56 kJ/mol while A ranged from $1.82 \times 10^6$ to $1.13 \times 10^6$ $\text{min}^{-1}$ at different pyrolysis conditions. The lowest E and A values were observed at higher pyrolysis temperature and lower heating rates. Under these conditions, the pyrolysis reaction becomes easier, as the required energy is smaller.

Table 7 lists the pyrolysis kinetic parameters of different types of organic (agro-industrial and agricultural residues) feedstocks, as reported in the international literature, and it compares them with the values of peach seeds pyrolysis found in this study.

**Table 7.** Comparison of peach seeds pyrolysis kinetic parameters with those of different biomass feedstocks reported in the literature.

| Biomass Residue | Process | Activation Energy (E, in kJ/mol) | Pre-Exponential Factor (A, in min$^{-1}$) | Ref. |
|---|---|---|---|---|
| Peach seeds | Slow pyrolysis | 30–60 | $10^3$–$10^4$ | Present study |
| Corn stalk | TG-DTG | 66.518 | $189.6 \times 10^3$ | [47] |
| Wheat straw | TG-DTG | 70.516 | $117.6 \times 10^6$ | [47] |
| Tree skin | TG-DGT | 77.316 | $157.2 \times 10^5$ | [47] |
| Peanut shell | TG-DGT | 84.47 | $88.2 \times 10^6$ | [47] |
| Cotton | TG-DGT | 200.9 | $175.8 \times 10^{14}$ | [47] |

As can be seen in Table 7, E of peach seeds pyrolysis was found in this study to be in the range of 30–60 kJ/mol, while the E for straw and stalk materials (wheat straw and corn stalk) was found by other researchers to be in the range of 60~80 KJ/mol, for woody materials (wood chip and peanut shell) in the range of 80~100 KJ/mol, and for cellulosic materials (cotton and filter paper) above 200 KJ/mol [47]. The reason for these differences can be attributed to the difference of the biochemical composition of the organic materials. Considering that agri-biomass is a mixture of cellulose, hemicellulose, lignin and a small quantity of extracts, their composition behaves differently under the same heating conditions of the pyrolysis process, with the main decomposition zone of cellulose to be at around 523–773 K resulting in little carbon, while the pyrolysis of lignin starts relatively earlier, resulting in relatively large quantity of carbon and hemicellulose showing decomposition at higher temperature between 500~600 K.

The higher cellulosic composition a feedstock has, the higher is the activation energy, as it is in the case of cotton feedstock (Table 7), because cotton is mainly composed of cellulosic polymers. E of pretreated peach seeds is the lowest because peach seeds residue contains a high amount of lignin and thus it decomposes generally easier, requiring a lower E (Table 7).

It can be argued that the differences of the kinetic parameters between the different biomasses in Table 7 can be attributed primarily to the different chemical composition of biomasses and ash content that implies different E requirements, but also to the transfer phenomena (heat transfer and mass transfer) happening in the reactor during the pyrolysis process.

The peach seeds kinetic parameters estimated in this study reveal that this residue is suitable for pyrolysis because it decomposes easily.

### 3.7. Estimation of the Combustion Index C

In order to assess the peach seeds as fuel, the estimation of the combustion indexes was made using TGA/DTG thermograms, as shown in Figure 6a,b. For the calculations, the combustion ignition temperature is needed, which is determined by the outset of weight drop in the thermogram. Furthermore, the burnout temperature is required, which is determined from the turning point to 99% conversion in the thermogram Weight loss is illustrated as a percentage of the initial weight of the sample, as a function of time (in minutes), and the weight loss rate in mg/min or % weight loss/min, respectively.

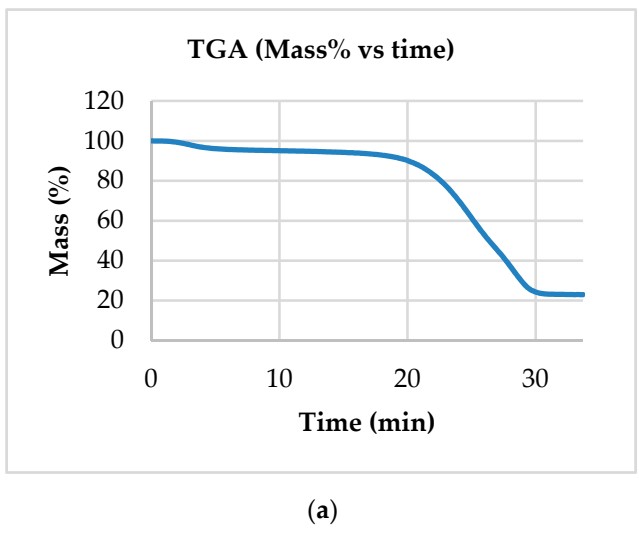
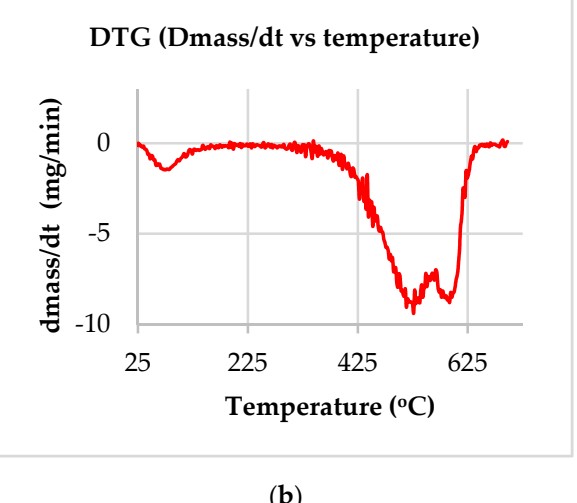

(**a**)  (**b**)

**Figure 6.** (**a**) Pyrolytic biochar weight loss diagram as a function of time (TGA, Air flow). (**b**) Biochar weight change rate diagram as a function of temperature (DTG, Air flow).

The rate of combustion is proportional to the kinetic constant, which in turn depends on temperature through the Arrhenius equation. Thus, the rate of combustion of a solid (or even liquid) fuel is expressed in terms of its temperature dependence.

From TG analysis, the weight loss curve of peach seeds materials under different combustion circumstances can be achieved. Approximately 10 mg of a sample of particle sizes of 250 μm was heated in a porcelain crucible up to 750 °C, at a rate of 10 °C/min to eliminate the effects of eventual side reactions and mass and heat transfer limitations. The sample weight loss and rate of weight loss were recorded continuously as functions of time or temperature. Typical peach seeds TG and DTG profiles are illustrated in Figure 5a,b.

Furthermore, to assess the pyrolytic biochar as a fuel, thermograms are presented in Figure 6a,b (TGA and DTG diagrams for the pyrolytic biochar).

For the calculation of the combustion index C, Figures 5 and 6 were used following the computational procedure described in Section 2.3. The values of the parameters, as well as the index's values, are presented in Table 8.

**Table 8.** Calculation of combustion index values.

| | Peach Seeds | | | | | |
|---|---|---|---|---|---|---|
| **Sample** | **Peach Seeds** | | | | **Pyrolytic Biochar** | |
| **TGA/ DTG** | **1st Peak, °C** | **1st Peak, °K** | **2nd Peak, °C** | **2nd Peak, K** | **Peak, °C** | **Peak, K** |
| $\left(\frac{dW}{dt}\right)_{max}$, %/min | 7.8 | 7.8 | 5.2 | 5.2 | 9 | 9 |
| $\left(\frac{dW}{dt}\right)_{max}$, %/min | 2.8 | 2.8 | 2.8 | 2.8 | 4.5 | 4.5 |
| $T_i$, °C/K | 195 | 468 | 475 | 748 | 400 | 673 |
| $T_e$, °C/K | 445 | 718 | 635 | 908 | 625 | 898 |
| Combustion index C | $1.9 \times 10^{-6}$ | $1.39 \times 10^{-7}$ | $1.02 \times 10^{-7}$ | $2.87 \times 10^{-8}$ | $4.05 \times 10^{-7}$ | $9.96 \times 10^{-8}$ |

In Table 8, the indexes were calculated by using the temperature both in Celsius (°C) and Kelvin (K). The reason was to comply with the international bibliography data on index values for other biomasses.

Figure 6b shows two peaks (which are the result of the change in gradients of the

weight drop curve in Figure 6a. The two peaks correspond to the combustion of volatiles and the combustion of the remaining char, respectively. The value of the combustion index of volatiles is by one order of magnitude higher than that of char, since on the one hand, the average and maximum rate (($dW/dt$)$_{mean}$ and ($dW/dt$)$_{max}$, respectively) are higher, and on the other hand, the ignition and extinguishing temperatures ($T_i$ and $T_e$, respectively) are lower. All these values contribute to the reduced value of the index for the second stage of combustion of the pretreated peach seeds (the combustion of its char). Better combustion quality of volatiles is expected, as gaseous fuels burn much better than solids, mainly due to better mixing conditions with air.

Figure 7a,b show the weight loss and weight loss rate in the TGA/DTG diagrams of the pyrolytic biochar produced from pretreated peach seeds. Based on these figures, the parameters and values of the combustion index were calculated and presented in Table 8. Biochar shows a fuel quality index lower than that of the combustion of volatiles of pretreated peach seeds, which is expected. The differences are due to the higher combustion rates, which are offset by the higher temperatures. The first contribute positively to the index, while the latter contribute negatively.

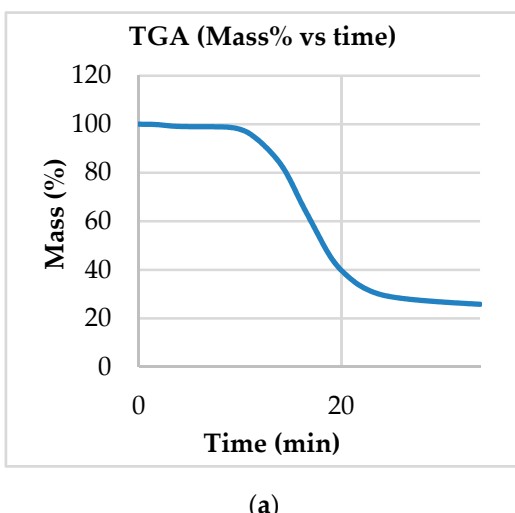 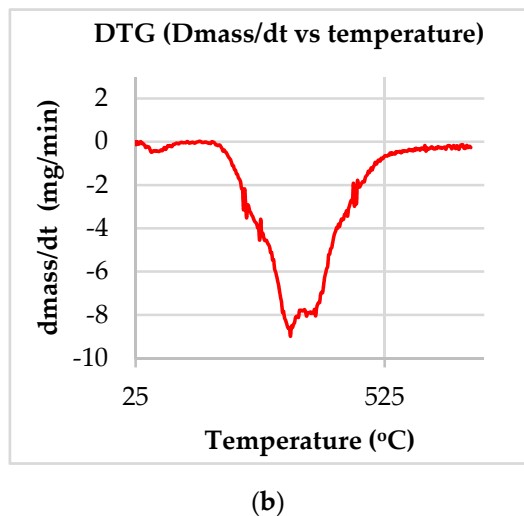

(**a**)  (**b**)

**Figure 7.** (**a**) Pretreated peach seeds weight loss diagram as a function of time (TGA, N$_2$ flow). (**b**) Pretreated peach seeds weight change rate diagram as a function of temperature (DTG, N$_2$ flow).

In conclusion, the pretreated peach seeds burns better than its char due to the volatiles it contains.

Table 9 is a comparative table between the combustion indexes of different biomasses.

**Table 9.** Combustion index C of pretreated peach seeds, biochar and of different biomasses under a heating rate of 20 °C/min.

| Sample | Peach Seeds, 1st Peak, | Peach Seeds Biochar | Peanut Shells [48] | Corn Stalk [49] | Rice Straw [49] | Rice Shells [49] | SAWDUST [49] |
|---|---|---|---|---|---|---|---|
| $\left(\frac{dW}{dt}\right)_{max}$, %/min | 7.8 | 9 | 4.21 | 13.49 | 11.56 | 9.49 | 14.86 |
| $\left(\frac{dW}{dt}\right)_{mean}$, %/min | 2.8 | 4.5 | 0.55 | 4.57 | 3.65 | 2.89 | 3.73 |
| $T_i$, °C/K | 195 | 400 | 236 °C | 261 | 257 | 276 | 293 |
| $T_e$, °C/K | 445 | 625 | 346 °C | 535 | 564 | 568 | 578 |
| Combustion index, C | $1.29 \times 10^{-6}$ | $4.05 \times 10^{-7}$ | $5.6 \times 10^{-8}$ | $1.69 \times 10^{-6}$ | $1.39 \times 10^{-7}$ | $6.34 \times 10^{-7}$ | $1.12 \times 10^{-6}$ |

As can be observed in Table 9, the pretreated peach seeds show close C index values with the corn stalk and sawdust, while rice straw, rice shells and peanut shells differ significantly downwards by one, one and two orders of magnitude, respectively. This is due to the slow combustion rates as evidenced by the TGA/DTG experiments (Table 9).

Therefore, pretreated peach seeds have a better burning quality than many other organic materials.

Estimation of the Pyrolysis Index D

From the TGA and DTG curves, pyrolysis of peach seeds can be divided into three stages. The first stage is the process of removal of surface water, that continued till 400 K. The second stage starts at 20 min later and it is the main reaction stage of pyrolysis of weight loss occurred, extending from 400 K to 525 K, and the weight loss velocity reaches the maximum at the curve peak. During the third stage starting at 30 min, the pyrolysis residue slowly decomposed, and the residue ratio tends to be constant at the end. The kinetic analysis focuses on the most severe stage of pyrolysis, that is, the main reaction period. During the TG-DTG analysis, the peach seeds sample with original mass ($m_0$) decomposes under program-controlled heating and with time (t) the sample changes to a mass (m) with a decomposition rate given by the Equation (22).

$$dx/dt = kf(x) \tag{22}$$

$$\text{The conversion rate is expressed as: } X = (m_0 - m)/(m_0 - m_\infty) \tag{23}$$

where $m_\infty$ is the biochar (residue mass) of the pyrolysis process at the end of pyrolysis process.

The volatile release index D and its parameters are calculated based on the TGA/DTG curves of Figure 7a,b.

Table 10 shows the volatile release index D (pyrolysis index) for the pretreated peach seeds of the present work. In the literature, the values of this specific index for biomass materials are very low. Indicatively, Table 10 presents the volatile release indexes for corn stalks and municipal waste (MSW, municipal solid wastes). A comparison of values between them of pretreated peach seeds pyrolysis with other biomasses shows that peach seeds can be pyrolyzed easily compared to corn stalks, since their values differ by two orders of magnitude, mainly due to the low cracking rates of the stalks. Even pyrolysis of municipal waste lacks behind pretreated peach seeds for about the same reason, although not all data for calculating the volatile release index are presented.

In conclusion, pretreated peach seeds are proving to be suitable for pyrolysis.

**Table 10.** Pyrolysis characteristics parameters indexes of pretreated peach seeds, and different types of biomass and volatile release index, D.

|  | Pretreated Peach Seeds, °C | Pretreated Peach Seeds, °K | Corn Stalk, [49] | MSW ($\times 10^{-7}$) [50] |
|---|---|---|---|---|
| $\frac{dw}{dt}_{max}$, %/min | 8.8 | 8.8 | 0.881 | 0.61 |
| $\frac{dw}{dt}_{mean}$, %/min | 2.19 | 2.19 | 0.148 | - |
| Starting temperature ($T_s$), °C, T | 195 | 468 | 183.3 | - |
| Peak temperature ($T_{max}$), °C, T | 420 | 693 | 334.4 | - |
| $\Delta T_{1/2}$ | 50 | 50 | 69.2 | 50–68 (*) |
| D | $4.7 \times 10^{-6}$ | $1.18 \times 10^{-6}$ | $3.0 \times 10^{-8}$ | 1.1–6.6 (*) |

(*) Range of values.

## 4. Conclusions

Peach seeds provided by a Greek fruit-processing company first underwent hexane extraction for the recovery of 36.8% wt. of oils contained, and the remaining residue was submitted to slow pyrolysis for energy and/or biochar recovery. Results of the pyrolysis experiments were used in this study for the pyrolysis apparent kinetics investigation.

In this study, exploring the kinetics of peach seeds pyrolysis in a systematic way implied not only experimental tests for all model reaction steps under a wide range of operating conditions, but also estimation of kinetic rate constants based on experimental data, in a series of computational steps that the used Coats–Redfern integral non-isothermal fitting model required.

The application of the Coats and Redfern model on the pyrolysis results revealed that the 3rd order chemical reaction best fits the whole process, while E and A values ranged from 23 to 56 kJ/mol and $1.82 \times 10^6$ to $1.13 \times 10^6$ min$^{-1}$, respectively, at different pyrolysis conditions. Regarding their dependence on the experimental conditions, it was found that an increase in pyrolysis temperature leads to a decrease in the activation energy (E) and the pre-exponential factor (A), while on the contrary, flow and heating rate increase lead to an increase in these kinetic parameters.

The estimation of indexes for the assessment of the suitability of peach seeds as a pyrolysis feedstock was performed by using TGA/DGTA. It was shown that the oil-extracted peach seeds are suitable feedstock for pyrolysis, having higher indexes than many other biomass residues.

This study provides insights for the apparent pyrolysis kinetic modeling of pre-valorized peach seeds, contributing to the knowledge required for the optimization of the pyrolysis process efficiency and scaling up, while proposing a waste cascade biorefinery in a circular bioeconomy solution for those food-industry residues. Valorization of the agri-food residues and waste constitutes an important sector within the agricultural production of Greece, holding a very important share in the domestic agricultural economy, with peach fruits production to constitute a vital agricultural sector. Valorization of peach seeds via oil extraction and consequent pyrolysis of the remaining solid could be a zero-waste, ecological, sustainable solution with material and energy savings in the circular bioeconomy journey that the country is undertaking. Under this concept, and for increasing the added value of peach seeds valorization, a cascade biorefinery approach is proposed that integrated oils extraction and pyrolysis.

**Author Contributions:** Conceptualization, A.Z.; methodology, G.S.; software, A.-I.A. and N.-C.K.; validation, A.-I.A. and N.-C.K.; investigation, A.-I.A. and N.-C.K.; resources, A.Z. and G.S.; data curation, A.-I.A. and N.-C.K.; writing—original draft preparation, A.-I.A. and N.-C.K.; writing—review and editing, A.Z.; visualization, A.-I.A. and N.-C.K.; supervision, A.Z.; project administration, A.Z. All authors have read and agreed to the published version of the manuscript.

**Funding:** This research received no external funding.

**Data Availability Statement:** Data is contained within the article.

**Conflicts of Interest:** The authors declare no conflict of interest.

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
