# Peer review of "Apparent Pyrolysis Kinetics and Index-Based Assessment of Pretreated Peach Seeds"

_processes, doi:10.3390/pr9060905_

Round 1

Reviewer 1 Report

Dear Authors, the paper "A kinetic study of Peach Seeds Pyrolysis and an Index-based Assessment" represents an interesting point of view to a well known topic. It may be considered as a contribution to the field, as the topic is very specific and accurate for only one raw material.

Though, the introduction section should be expanded, the reference list could be updated, to make the article more interesting for a broad reader. Mentioning different materail processed within the same procedure would be of great value.

The materials and Methods are well described, however it should be mentioned, that some methods are described in literrature, if applicable. The humidity and time of storage should be mentioned, as they may affect the results.

The equations 1-9 are academic and if they are not an original result, they should be reffered.

Figure 2 is a reprint? If so, the authors should coinsider to cite it instead of presenting ot another time.

The mathematical analysis is very advanced, however the presented results are given without at least OD, therefore it seems that control is missing. Also the range of experiment must be defined. The discussion is carried carefully and is accurate, the conclusions are supported by the results.

Author Response

Reply to Comments and Suggestions for Authors

Processes-1225588R1

REPLY TO REVIEWER #1

Comment 1:    Dear Authors, the paper "A kinetic study of Peach Seeds Pyrolysis and an Index-based Assessment" represents an interesting point of view to a well-known topic. It may be considered as a contribution to the field, as the topic is very specific and accurate for only one raw material. Though, the introduction section should be expanded, the reference list could be updated, to make the article more interesting for a broad reader. Mentioning different material processed within the same procedure would be of great value.

Reply 1:           We expanded slightly the introduction because the theory on kinetic modelling is included in the theoretical part (materials and methods). We revised the scope and objective of the paper as well.  We also added in the results part data from other studies to compare with the results of this study.

Comment 2:    The materials and methods are well described, however it should be mentioned, that some methods are described in the literature, if applicable. The humidity and time of storage should be mentioned, as they may affect the results.

Reply:              The sample we use is dry. We added in paragraph 2.1 the sentence:

    An initial sample mass of approx. of spherical shaped granules with a particle size varying between 500-850 μm was evenly arranged in a metal plate and placed in the centre of the reactor. The initial sample moisture did not exceed 10% wt./wt.

Comment 3:    The equations 1-9 are academic and if they are not an original result, they should be referred.

Reply 3:          They are cited as references [8], [9], [10], [11] appropriately.

Comment 4:    Figure 2 is a reprint? If so, the authors should consider citing it instead of presenting of another time.

Reply 4:           It is a reprint and it is cited as [13].

Comment 5:    The mathematical analysis is very advanced; however the presented results are given without at least OD, therefore it seems that control is missing. Also, the range of the experiment must be defined.

Reply 5:           The experimental results are added in Paragraph 3.1

Comment 6:    The discussion is carried carefully and is accurate, the conclusions are supported by the results.

Reply 6:           Thank you very much.

Reviewer 2 Report

The authors establish a 3rd order Pyrolsis reaction using Coates and Redfern software. The authors did a good job explaining their experimental methods. The paper reads a bit like a dissertation chapter.  Can the equations be paired down to the most important ones? Some of the others ones put in a Supporting information.  While I think this will be important to some readers, it distracts from the overall content, a supporting information document showing all the equations may help and only highlighting the most important.  

  1. The statements in the introduction need references. 
  2. The transition to paragraph on line 37 seems abrupt.  A line is missing to make that transition smoother.
  3. Under materials, is it mortar supposed to be mortar and pestle.
  4. Chemicals used such as hexane should have where they are purchased from
  5. Make figure 6 the same size as above
  6. Your graph figures are publication quality.  I.e they are what excel gives you. They should look better for the paper
  7. All your graphs are different sizes
  8. Do you have any composition studies of what's in the seeds? Or what products are burning off. Some TGA have a mass spec attached to them
  9. On line 497 it says that one burns better because volatile content, could water content play a role

Author Response

REPLY TO REVIEWER #2

Comment 1:    The authors establish a 3rd order Pyrolsis reaction using Coates and Redfern software. The authors did a good job explaining their experimental methods. The paper reads a bit like a dissertation chapter.  Can the equations be paired down to the most important ones? Some of the other’s ones put in Supporting information.  While I think this will be important to some readers, it distracts from the overall content, a supporting information document showing all the equations may help and only highlighting the most important.

Reply 1:           We reformed the theoretical part and added experimental results as well.

Comment 2:    The statements in the introduction need references.

Reply 2:           References cited in the introduction.

Comment 3:    The transition to the paragraph on line 37 seems abrupt.  A line is missing to make that transition smoother.

Reply 3:           The transition is smoother now.

Comment 4:    Under materials, is it mortar supposed to be mortar and pestle.

Reply 4:           The word ‘Mortar’ was deleted.

Comment 5:    Chemicals used such as hexane should have where they are purchased from.

Reply 5:           Hexane is a MERCK product

Comment 6:    Make figure 6 the same size as above

Reply 6:           We made it

Comment 7:    Your graph figures are publication quality.  I.e they are what excel gives you. They should look better for the paper.

Reply 7:           We tried to upgrade them.

Comment 8:    All your graphs are different sizes

Reply 8:           We tried to give them the same size but all cannot be the same or appear the same

Comment 9:    Do you have any composition studies of what's in the seeds? Or what products are burning off? Some TGA have a mass spec attached to them

Reply 9:           We added the composition of the peach seeds are requested.

Comment 10:  On line 497 it says that one burns better because volatile content, could water content play a role

Reply 10:         Thank you for the comment. All of them have the same amount of moisture (around 10%).

Round 2

Reviewer 2 Report

The authors have responded to my comments.

Author Response

Reply to EDITOR

I read the revised manuscript and I can see that an effort has been carried out by the Authors to take into proper account the comments from the Reviewers and from my side. However, before to be accepted for publication in the Special Issue on Biomass Pyrolysis Kinetics I still have an additional request concerning an important aspect, as follows.

General Comment:         I still have some doubt about the mathematical method applied for the kinetic evaluation. The authors state that the weight loss measurements cannot be considered isothermal, and it is right given the not sufficiently high heating rate (nominal value of 100C/min), also considering the quite large initial sample mass (about 1g). However, the analytical treatment of the thermogravimetric data seems to be based on isothermal data. In particular, I wonder what do the temperatures reported in the Arrhenius plot (Figure 3 represent? In the case of isothermal data, a point on the Arrhenius plot is obtained for each temperature at which the weight loss has been measured. But in this case? This point should be clarified.

Reply 1:  Thank you for this comment.  This helped us to clarify better what we have done. As we now stated in the abstract and in the scope, this study aims at the apparent kinetic investigation of peach seeds slow pyrolysis that performed at a lab unit, under inert atmospheric conditions at different temperatures (475-785 oC), heating rates (100-250 oC/min) and N2 flow rates (25-200 cc/min). Prior to pyrolysis, peach seeds have submitted to hexane extraction for the recovery of 36.8% wt. of the contained oils. Determination of the specific rate constant (k) and activation energy (Ea) for each considered reaction was made by using the Coats–Redfern integral non-isothermal fitting model that requires an assumption of the reaction order (n).

The TGA used only for the estimation of combustion/pyrolysis indexes made to assess the suitability of peach seeds as a fuel, using isothermal thermogravimetric analyses (TGA).        

Minor comments

 Comment 1:      Minor suggestions: As I already pointed out in my comments about the original version of the manuscript, some steps included in the description of the experimental procedure for the pyrolysis tests are very trivial (i.e. " Open of a new Genie software tab for monitoring the experiment "). As Reviewer 2 states, this part seems to write more like a " dissertation chapter" rather than a scientific paper. So, I suggest summarizing this explanation keeping in only the actually "important" steps of the experimental procedures. As Reviewer 1 wrote, a similar observation can be also made for the description of the mathematical methods applied for the apparent kinetics estimation and for the combustion index evaluation.

Reply 2:  We summarized the description of the trivial experimental procedure for the pyrolysis tests, as suggested (lines 187-202). We also summarized the description of the mathematical methods applied for the apparent kinetics estimation and for the combustion index evaluation (lines 420-424).

Comment 2:       The same abscissa, e.g. the temperature, should be used for the Figure 6  and 7 (the mass curve in Fig.6 is currently reported as a function of time). Actually, these two figures (6 and 7) could be combined in only one, but this is left to the Authors' likes.

Reply 2: We combine Fig.6 and Fig.7 in one (Fig.6a,b), Fig.8 and Fig.9 in one (Fig.7a.b), and Fig.10 and Fig 11., in (Fig 8a,b) as suggested.

Comment 3:  In eq. 2 the letter "A" is used for the pre-exponential factor of the Arrhenius expression, while k0 is used in the following. Please, use the same letter.

Reply 3:    It was corrected, Thank you!

Reviewer 2:                                                                                                                  Comment 1:       Provide more references in the introduction.

Reply 1:              We extended the introduction by adding a paragraph on the discussion the kinetic modelling (line 77- line 117) via 27 new references (see also the list of references)
